# Constrained Online Convex Optimization with Polyak Feasibility Steps

Spencer Hutchinson [1]   Mahnoosh Alizadeh [1]

## Abstract

In this work, we study online convex optimization with a fixed constraint function $g : \mathbb{R}^d \to \mathbb{R}$. Prior work on this problem has shown $\mathcal{O}(\sqrt{T})$ regret and *cumulative* constraint satisfaction $\sum_{t=1}^{T} g(\mathbf{x}_t) \leq 0$, while only accessing the constraint value and subgradient at the played actions $g(\mathbf{x}_t), \partial g(\mathbf{x}_t)$. Using the same constraint information, we show a stronger guarantee of *anytime* constraint satisfaction $g(\mathbf{x}_t) \leq 0 \; \forall t \in [T]$, and matching $\mathcal{O}(\sqrt{T})$ regret guarantees. These contributions are thanks to our approach of using *Polyak feasibility steps* to ensure constraint satisfaction, without sacrificing regret. Specifically, after each step of online gradient descent, our algorithm applies a subgradient descent step on the constraint function where the step-size is chosen according to the celebrated Polyak step-size. We further validate this approach with numerical experiments.

## 1. Introduction

We study the problem of online convex optimization (OCO) where, in each round $t = 1, 2, ..., T$, a player chooses an action $\mathbf{x}_t$ from an action set $\mathcal{X} \subseteq \mathbb{R}^d$ and then suffers the cost $f_t(\mathbf{x}_t)$ according to an adversarially-chosen convex function $f_t$ (Zinkevich, 2003). The player's goal is to minimize the regret with respect to the best single action in hindsight,

$$\text{Reg}_T := \sum_{t=1}^{T} f_t(\mathbf{x}_t) - \min_{\mathbf{x} \in \mathcal{X}} \sum_{t=1}^{T} f_t(\mathbf{x}).$$

The OCO problem has emerged as a fundamental setting for various machine learning domains, such as stochastic optimization (Cesa-Bianchi et al., 2004), non-convex opti-

mization (Cutkosky et al., 2023), and online control (Agarwal et al., 2019). Furthermore, the OCO problem is directly relevant to various real-world settings, including online advertising (McMahan et al., 2013), internet of things (Chen & Giannakis, 2018), and healthcare (Tewari & Murphy, 2017).

Despite the significance of the OCO problem, classical methods for OCO, such as online gradient descent (OGD) (Zinkevich, 2003) and regularized follow the leader (RFTL) (Shalev-Shwartz & Singer, 2007), require orthogonal projections (or similarly costly operations) to ensure that the chosen actions are feasible. Such operations are prohibitively expensive in many applications, particularly when the action set $\mathcal{X}$ has a complicated structure. Motivated by the high computation cost of classical methods, Mahdavi et al. (2012) introduced the problem of *OCO with long-term constraints*, where the player is *not* required to ensure that the actions are feasible, but instead aims to satisfy the constraints *cumulatively*.

In particular, Mahdavi et al. (2012) considered the action set to be represented by the sub-level set of a constraint function, i.e. $\mathcal{X} = \{x \in \mathbb{R}^d : g(x) \leq 0\}$ for convex[1] $g : \mathbb{R}^d \to \mathbb{R}$. In this setting, Mahdavi et al. (2012) gave an algorithm that enjoys $\mathcal{O}(T^{3/4})$ regret and ensures cumulative constraint satisfaction $\sum_{t=1}^{T} g(\mathbf{x}_t) \leq 0$, while only observing a subgradient and function value at the played actions, i.e. $g_t = g(\mathbf{x}_t), \mathbf{s}_t \in \partial g(\mathbf{x}_t)$. This algorithm uses a *primal-dual* approach, in which it updates sequences of actions (primal variables) and constraint penalties (dual variables) to simultaneously minimize the regret and the cumulative constraint value. Primal-dual approaches have continued to be successful for this problem, as more recent algorithms of this type have been shown to enjoy $\mathcal{O}(\sqrt{T})$ regret with the same constraint guarantees and constraint feedback (Yu et al., 2017).

In this work, we take a new approach to this problem by using what we call *Polyak feasibility steps*, which are subgradient descent steps that are taken with respect to the constraint function and use the Polyak stepsize. This approach is motivated by the fact that the Polyak stepsize is known to be effective in unconstrained convex optimization (Polyak,

---

[1]Department of Electrical and Computer Engineering, University of California-Santa Barbara, Santa Barbara, California, USA. Correspondence to: Spencer Hutchinson <shutchinson@ucsb.edu>, Mahnoosh Alizadeh <alizadeh@ucsb.edu>.

*Proceedings of the 42$^{nd}$ International Conference on Machine Learning*, Vancouver, Canada. PMLR 267, 2025. Copyright 2025 by the author(s).

---

[1]Since the constraint function $g : \mathbb{R}^d \to \mathbb{R}$ is *not* assumed to be differentiable, this setting can handle multiple constraints $g_1, ..., g_n$ by defining $g(\mathbf{x}) := \max_{i \in [n]} g_i(\mathbf{x})$.

*Table 1.* State-of-the-art algorithms for OCO with a non-smooth functional constraint $\mathcal{X} = \{\mathbf{x} \in \mathbb{R}^d : g(\mathbf{x}) \leq 0\}$ and first-order feedback $g_t = g(\mathbf{x}_t), \mathbf{s}_t \in \partial g(\mathbf{x}_t)$. Our results are highlighted with gray. The results marked with * have been extended with Assumption 1 and Theorem 7 in Mahdavi et al. (2012) as we show in Appendix A. The column "Known Strictly-Feasible Point?" is marked "Yes" if the algorithm requires knowledge of an $\mathbf{x} \in \mathbb{R}^d$ such that $g(\mathbf{x}) < 0$, and "No" if not.

| Reference | Regret | Constraint Guarantee | Known Strictly-Feasible Point? |
|---|---|---|---|
| Mahdavi et al. (2012) | $\mathcal{O}(T^{3/4})$ | $\sum_{t=1}^{T} g(\mathbf{x}_t) \leq 0$ | No |
| Jenatton et al. (2016) | $\mathcal{O}(T^{2/3})$ | $\sum_{t=1}^{T} g(\mathbf{x}_t) \leq 0$ | No |
| Yuan & Lamperski (2018)* | $\mathcal{O}(T^{2/3})$ | $\sum_{t=1}^{T} g(\mathbf{x}_t) \leq 0$ | No |
| Yu et al. (2017)* | $\mathcal{O}(\sqrt{T})$ | $\sum_{t=1}^{T} g(\mathbf{x}_t) \leq 0$ | No |
| Corollary 1 | $\mathcal{O}(\sqrt{T})$ | $g(\mathbf{x}_t) \leq 0, \; \forall t \in [T]$ | Yes |
| Corollary 2 | $\mathcal{O}(\sqrt{T})$ | $g(\mathbf{x}_t) \leq 0, \; \forall t \geq t_0 = \mathcal{O}(\log(T))$ | No |
| Corollary 2 | $\mathcal{O}(\sqrt{T})$ | $\sum_{t=1}^{T} g(\mathbf{x}_t) \leq 0$ | No |

1969). As such, we expect it to be similarly effective when applied to the constraint function in OCO. Accordingly, we design an algorithm that alternates between gradient descent steps with respect to the cost function, and Polyak feasibility steps. This algorithm differs from primal-dual approaches in that it maintains a *single* sequence of iterates (versus the *two* sequences used by primal-dual algorithms).

We find that our approach enjoys $\mathcal{O}(\sqrt{T})$ regret and *anytime constraint satisfaction* $g(\mathbf{x}_t) \leq 0 \; \forall t$, while still only observing a subgradient and function value of the constraint at the played actions. Unlike prior methods for constrained OCO, which often trade feasibility for efficiency, our approach enjoys both feasibility *and* efficiency. Indeed, our algorithm maintains constraint satisfaction in all rounds, while using the same constraint feedback as prior work and avoiding the use of projections. Furthermore, our approach is relevant to safety-critical applications, in which constraint satisfaction is paramount and there is often only limited constraint information available.

Our complete results are shown alongside prior work in Table 1. As presented in this table, prior work has shown *cumulative* constraint satisfaction $\sum_{t=1}^{T} g(\mathbf{x}_t) \leq 0$ and regret bounds as tight as $\mathcal{O}(\sqrt{T})$ (Mahdavi et al., 2012; Jenatton et al., 2016; Yuan & Lamperski, 2018; Yu et al., 2017). We show stronger *anytime* constraint satisfaction $g(\mathbf{x}_t) \leq 0 \; \forall t \in [T]$, while also guaranteeing $\mathcal{O}(\sqrt{T})$ regret. However, unlike prior work, these guarantees require that there is a known point $\mathbf{x} \in \mathbb{R}^d$ that is strictly-feasible $g(\mathbf{x}) < 0$. Nonetheless, we show that when a strictly-feasible point is *not* known, then we can guarantee constraint satisfaction after $\mathcal{O}(\log(T))$ rounds, as well as cumulative constraint satisfaction.

Also, note that Table 1 only includes algorithms that access the constraint function via the constraint value and subgradient at the played actions $g_t = g(\mathbf{x}_t), \mathbf{s}_t \in \partial g(\mathbf{x}_t)$. Therefore, it does not include the line of literature in con-

strained OCO that solves a convex optimization problem with the constraint function in each round, e.g. (Yu & Neely, 2020; Yi et al., 2021; Guo et al., 2022). We discuss these related works in more detail in the following section.

## 1.1. Related Work

In this section, we discuss related work on constrained OCO, projection-free OCO, and constrained optimization.

### 1.1.1. CONSTRAINED OCO

Mahdavi et al. (2012) studied OCO with a fixed convex constraint function $g$, and gave an algorithm with $\mathcal{O}(\sqrt{T})$ regret and $\mathcal{O}(T^{3/4})$ cumulative violation $\sum_{t=1}^{T} g(\mathbf{x}_t)$ that used only first-order constraint feedback, $g_t = g(\mathbf{x}_t), \mathbf{s}_t \in \partial g(\mathbf{x}_t)$. This result was then generalized to $\mathcal{O}(T^{\max(\beta, 1-\beta)})$ regret and $\mathcal{O}(T^{1-\beta/2})$ cumulative violation for any $\beta \in (0, 1)$ by Jenatton et al. (2016). The same bounds were shown for a stronger notion of constraint violation $\sum_{t=1}^{T}[g(x_t)]_+$ by Yuan & Lamperski (2018), who also guaranteed that $\sum_{t=1}^{T}([g(x_t)]_+)^2 = \mathcal{O}(T^{1-\beta})$. Finally, Yu et al. (2017) showed $\mathcal{O}(\sqrt{T})$ regret and $\mathcal{O}(\sqrt{T})$ cumulative violation in the same setting. With the additional assumption that the constraint gradient is lower bounded near the constraint boundary (Assumption 1 in Mahdavi et al. (2012)), the aforementioned results can be extended to guarantee no cumulative violation $\sum_{t=1}^{T} g(\mathbf{x}_t) \leq 0$ as stated in Table 1. We use the same assumption to show constraint satisfaction for all rounds $g(x_t) \leq 0 \; \forall t$ provided that there is a known strictly-feasible point. Furthermore, the aforementioned works use primal-dual algorithms, which are fundamentally different from our approach of Polyak feasibility steps. In particular, our approach uses the Polyak step-size and a *single* sequence of iterates, while primal-dual algorithms use two iterate sequences that are linked via the cost and constraint functions.

There is also a line of literature on OCO with constraints

that solves an optimization problem involving the constraint function in each round, e.g. (Yu & Neely, 2020; Yi et al., 2021; 2022; Guo et al., 2022). This differs from our algorithm and those compared in Table 1, which only access the constraint with first-order feedback at the played actions. We note that solving an optimization problem with the constraint function in each round can introduce significant computational cost.

Lastly, we point out that a related line of literature considers OCO with time-varying constraints, e.g. (Neely & Yu, 2017; Yu et al., 2017; Liakopoulos et al., 2019; Castiglioni et al., 2022; Guo et al., 2022; Kolev et al., 2023). This generalizes the fixed constraints that we consider. Nonetheless, to our knowledge, none of these works improve on the guarantees shown in Table 1 for the case of non-smooth constraints, and first-order feedback. We also point out that Kolev et al. (2023) gives an algorithm that uses first-order feedback and enjoys bounds on the violation in each round, i.e. $g(x_t) \leq \mathcal{O}(\frac{1}{\sqrt{t}})$. However, this work requires that constraint functions are smooth (i.e. have Lipschitz gradients), making it distinct from our work and those considered in Table 1. Furthermore, Kolev et al. (2023) uses a projection on to a polytope in each round, which introduces additional computational cost.

### 1.1.2. PROJECTION-FREE OCO

In parallel to the literature on constrained OCO, there is a line of literature that considers projection-free OCO where the feasible set is not treated as the sub-level set of a function, but rather defined as an arbitrary convex set. Hazan & Kale (2012) initiated this literature by giving an algorithm that accesses the feasible set via a linear optimization oracle (LOO) instead of using projections. This is advantageous because the LOO is often computationally cheaper than the projection. LOO-based algorithms have received a significant amount of attention, e.g. (Hazan & Kale, 2012; Chen et al., 2019; Garber & Kretzu, 2020; 2022; Wang et al., 2024; Garber & Kretzu, 2024). The state-of-the-art for LOO-based algorithms with general convex costs and general convex feasible sets is $\mathcal{O}(T^{3/4})$ regret and 1 oracle call per a round (Hazan & Kale, 2012). Although our methods give smaller $\mathcal{O}(\sqrt{T})$ regret, we note that the first-order feedback that we use is generally incomparable to the LOO oracle in terms of computationally complexity. Indeed, the first-order feedback is cheaper to compute for some constraint functions, while the LOO is cheaper to compute for other constraint functions.[2]

There is also a growing body of literature that uses the membership oracle (MO) or separation oracle (SO) to access the feasible set, e.g. (Levy & Krause, 2019; Garber & Kretzu, 2022; Mhammedi, 2022; Lu et al., 2023; Hu et al., 2023; Mhammedi, 2024). The MO and SO are defined as follows. Given a query point $\mathbf{x} \in \mathbb{R}^d$, the MO specifies whether or not $\mathbf{x}$ is in the feasible set, while the SO returns a hyperplane that separates $\mathbf{x}$ from the feasible set (if $\mathbf{x}$ is not in the feasible set). If the feasible set is the sub-level set of a constraint function $g$ (as we consider), then the MO can be constructed by checking if $g(\mathbf{x}) > 0$, and the SO can be constructed using the first-order information at $\mathbf{x}$, i.e. $g(\mathbf{x}), \partial g(\mathbf{x})$.[3] We use first-order information only at the played actions, which can therefore only be used to construct a separation oracle and membership oracle *at the played action*. This is distinct from existing MO and SO-based algorithms, which query the SO and MO at arbitrary points (not just the played actions). Furthermore, existing MO-based and SO-based algorithms require *multiple* oracle calls per a round. Specifically, existing MO-based algorithms use $\mathcal{O}(d \log(T))$ oracle calls per a round (Lu et al., 2023; Mhammedi, 2022), and existing SO-based algorithms use $\mathcal{O}(\log(T))$ oracle calls per a round (Mhammedi, 2022; 2024) or $\mathcal{O}(\kappa)$ oracle calls per a round (Garber & Kretzu, 2022). Note that we state these bounds for unrestricted $T$ and use $\kappa$ to refer to the eccentricity of the feasible set, i.e. $\kappa = R/r$ with $r\mathbb{B} \subseteq \mathcal{X} \subseteq R\mathbb{B}$, which can be arbitrarily large. The fact that our algorithm only requires 1 oracle call per a round can result in significant performance advantages over these methods, particularly when the constraint is costly to evaluate. Furthermore, our approach is applicable to settings where there is only *local* constraint information available.

### 1.1.3. CONSTRAINED OPTIMIZATION

Our approach is inspired by a line of literature in constrained (offline) optimization that uses the Polyak step-size to ensure convergence to the feasible set, e.g. (Polyak, 2001; Nedić, 2011; Nedić & Necoara, 2019; Necoara & Singh, 2022). However, we point out two key difficulties that arise in the OCO setting: (a) the suboptimality gap of the iterates $f_t(\mathbf{x}_t) - f_t(\mathbf{x}^\star)$ is *not* guaranteed to be non-negative, and (b) the constraint feedback is at the played action $\mathbf{x}_t$ and not at the "intermediate" iterate (labeled $\mathbf{y}_t$ in Algorithm 1). Challenge (a) is particularly difficult to handle because the analysis approach used by Nedić (2011) (and following works) relies on the suboptimality gap of the it-

---

[2]An example of this, pointed out by Garber & Kretzu (2022), is the difference in computational complexity for first-order information and LOO for the nuclear norm ball $\mathbb{B}_*$ and spectral norm ball $\mathbb{B}_2$ in the space of matrices. Computing first-order information for $\mathbb{B}_*$ and $\mathbb{B}_2$ is at worst a full-rank SVD (which is expensive) and rank-one SVD (which is cheap), respectively. For the LOO,

the opposite is true in the $\mathbb{B}_*$ requires a rank-one SVD and $\mathbb{B}_2$ a full-rank SVD.

[3]Given the query point $\mathbf{x}$ and first-order information at this point $g = g(\mathbf{x}), \mathbf{s} \in \partial g(\mathbf{x})$, it follows from convexity that $\{\mathbf{y} \in \mathbb{R}^d : g + \mathbf{s}^\top(\mathbf{y} - \mathbf{x}) \leq 0\}$ is a separating hyperplane w.r.t. $\mathcal{X} = \{\mathbf{x} \in \mathbb{R}^d : g(\mathbf{x}) \leq 0\}$.

erates being non-negative. As a result, we require a new analysis approach. However, our approach does suffer a larger dependence on the problem parameters (such as the subgradient bound $G_g$), which can be seen as the "cost" of the adversarial online setting. To handle challenge (b), we use the first-order approximation of the cost function $g_t + \mathbf{s}_t^\top (\mathbf{y}_t - \mathbf{x}_t)$ in the Polyak step-size instead of the true cost function $g(\mathbf{y}_t)$. This first-order approximation avoids the need for constraint information at the intermediate iterate $\mathbf{y}_t$, while maintaining the advantageous properties of the Polyak step-size (see Section 3.4 for the details). Such a first-order approximation is also advantageous in the offline setup as seen in the recent work of Singh & Necoara (2025), but the analysis in Singh & Necoara (2025) does not extend to the OCO setting because it requires that the suboptimality gap is non-negative (i.e. $f(x_t) - f(x^\star) \geq 0$) and that the cost functions have Lipschitz gradients. Related algorithm designs have also been considered in the literature on constrained variational inequalities (Zhang et al., 2025), although under the assumption that the constraint functions have Lipschitz gradients (which does not hold in our setting). We further discuss the differences between our algorithm and the algorithm in Zhang et al. (2025) in Appendix B.

## 1.2. Notation

We use $\mathcal{O}(\cdot)$ for big-O notation and $\tilde{\mathcal{O}}(\cdot)$ for the same ignoring log factors. The 2-norm is denoted by $\|\cdot\|$ and the 2-norm ball is denoted by $\mathbb{B} = \{\mathbf{x} \in \mathbb{R}^d : \|\mathbf{x}\| \leq 1\}$. Given a natural number $n$, we use the notation $[n] := \{1, 2, ..., n\}$. The transpose of a matrix $M$ is denoted $M^\top$. A vector of ones and zeros is denoted $\mathbf{1}$ and $\mathbf{0}$, respectively. For a given $x \in \mathbb{R}$, we use the notation $[x]_+ = \max(x, 0)$. Lastly, for a point $\mathbf{z} \in \mathbb{R}^d$ and closed convex set $\mathcal{Y} \subseteq \mathbb{R}^d$, we use the notation $\text{dist}(\mathbf{z}, \mathcal{Y}) = \min_{\mathbf{y} \in \mathcal{Y}} \|\mathbf{z} - \mathbf{y}\|$ and $\Pi_\mathcal{Y}(\mathbf{z}) = \arg\min_{\mathbf{y} \in \mathcal{Y}} \|\mathbf{z} - \mathbf{y}\|$.

## 1.3. Overview

We specify the problem of OCO with constraints in Section 2. Then, in Section 3, we give an algorithm for this problem that uses our approach of *Polyak feasibility steps*. In particular, Section 3.1 gives the description of this algorithm, Section 3.2 gives the guarantees of $\mathcal{O}(\sqrt{T})$ regret and *anytime* constraint satisfaction $g(\mathbf{x}_t) \leq 0$, and Section 3.3 and 3.4 give the regret analysis and feasibility analysis, respectively. Lastly, we give simulation results in Section 4 that demonstrate the functionality of our algorithm.

## 2. Problem Setup

We study the problem of online convex optimization with functional constraints. In the following, we first describe online convex optimization generally and then specify the functional constraints.

**Online Convex Optimization** Online convex optimization (OCO) is a repeated game between a player and an adversary that is played over $T$ rounds. In each round $t \in [T]$, the player chooses an action $\mathbf{x}_t$ from a convex action set $\mathcal{X} \subseteq \mathbb{R}^d$, and then the adversary chooses a convex function $f_t : \mathbb{R}^d \to \mathbb{R}$. The player aims to minimize the cumulative regret,

$$\text{Reg}_T := \sum_{t=1}^T f_t(\mathbf{x}_t) - \min_{\mathbf{x} \in \mathcal{X}} \sum_{t=1}^T f_t(\mathbf{x}),$$

We will use the standard assumptions that the action set is bounded and that the cost functions have bounded gradients. These are stated precisely in the following.

**Assumption 1.** *There exists positive real $R$ such that $\mathcal{X} \subseteq R\mathbb{B}$.*

**Assumption 2.** *There exists positive real $G_f$ such that $\|\nabla f_t(\mathbf{x})\| \leq G_f$ for all $\mathbf{x} \in R\mathbb{B}$ and $t \in [T]$.*[4]

**Functional Constraints** Following Mahdavi et al. (2012), we study the setting where the action set is defined by a functional inequality constraint $\mathcal{X} = \{\mathbf{x} \in \mathbb{R}^d : g(\mathbf{x}) \leq 0\}$, with $g : \mathbb{R}^d \to \mathbb{R}$ being non-smooth and convex. In the following, we assume that the constraint function has bounded subgradients (Assumption 3) and that the subgradient norm is lower-bounded near the boundary (Assumption 4). These assumptions are also used by Mahdavi et al. (2012) and Jenatton et al. (2016). We also note in Remark 1 that Assumption 4 is implied by Slater's condition (i.e. the existence of a strictly-feasible point) provided that the other assumptions hold. Lastly, note that since the constraint function is non-smooth, this setting can be extended to multiple constraints $g_1, .., g_m$ by taking $g(\mathbf{x}) = \max_{i \in [m]} g_i(\mathbf{x})$.

**Assumption 3.** *There exists a positive real $G_g$ such that, for all $\mathbf{x} \in R\mathbb{B}$, it holds that $\|\partial g(\mathbf{x})\| \leq G_g$.*

**Assumption 4.** *There exists positive reals $\sigma, \epsilon$ such that $\mathcal{X}' = \{\mathbf{x} \in \mathbb{R}^d : g(\mathbf{x}) = -\epsilon\}$ is nonempty and $\|\partial g(\mathbf{x})\| \geq \sigma$ for all $\mathbf{x} \in \mathcal{X}'$.*

**Remark 1.** *Assumption 4 holds if Assumption 1 holds and there exists a strictly-feasible point (i.e. Slater's condition). We show this in Appendix C.1.*

## 3. Algorithm

In this section, we give Algorithm 1, which tackles OCO with functional constraints using our approach of Polyak feasibility steps. Notably, Algorithm 1 only uses *one* constraint query in each round, at the played action $g_t = g(\mathbf{x}_t)$, $\mathbf{s}_t \in$

---

[4]This assumption can be weakened slightly for our guarantees of anytime constraint satisfaction (i.e. Corollary 1). Specifically, we only need that $\|\nabla f_t(\mathbf{x})\| \leq G_f$ for all $\mathbf{x} \in \mathcal{X}$ (rather than all $\mathbf{x} \in R\mathbb{B}$). We discuss this later in Remark 2.

$\partial g(\mathbf{x}_t)$, and therefore uses the *exact same* feedback as Mahdavi et al. (2012). Despite this limited feedback, we will show that Algorithm 1 ensures constraint satisfaction for all rounds, i.e. $g(\mathbf{x}_t) \leq 0$ for all $t \in [T]$.

### 3.1. Description

At a high-level, Algorithm 1 operates by alternating between gradient descent steps (line 4) and Polyak feasibility steps (line 5). We discuss the key ingredients of our Polyak feasibility steps in the following.

**Polyak Step-size**  The design of our Polyak feasibility step is motivated by the classical Polyak step-size (Polyak, 1969). In unconstrained convex optimization, using subgradient descent with the Polyak step-size is known to be optimal (Boyd et al., 2003), and therefore it is a natural choice to ensure strong feasibility guarantees when applied to the constraint function. This classical step-size uses the function value and subgradient at the current iterate to approximate the optimal step-size in each update. In our setting, this would require the constraint function value at the "intermediate iterate" $\mathbf{y}_t$, which is not available. We address this next.

**First-order Approximation**  As discussed previously, the classical Polyak step-size cannot immediately be applied to our setting because it would require the constraint function value at the intermediate iterate $\mathbf{y}_t$. Although this is not known in our setting, we do have constraint information at the played action $\mathbf{x}_t$, which should not be too far from the intermediate iterate $\mathbf{y}_t$. Therefore, we use the constraint information at $\mathbf{x}_t$ to construct a first-order approximation of the constraint at $\mathbf{y}_t$,

$$g(\mathbf{y}_t) \approx g_t + \mathbf{s}_t^\top (\mathbf{y}_t - \mathbf{x}_t) \in g(\mathbf{x}_t) + \partial g(\mathbf{x}_t)^\top (\mathbf{y}_t - \mathbf{x}_t). \quad (1)$$

As we will show in the analysis, this first-order approximation is sufficient to maintain the advantageous properties of the Polyak step-size.

**Constraint Tightening**  Another difficulty that arises in our setting is that the gradient descent step in line 4 might push the sequences of actions out of the feasible set. Indeed, the cost functions are chosen adversarially and therefore we have no guarantees about the direction of the gradient in each round. Therefore, to ensure that the actions are feasible, we use a tightened version of the constraint function $g(\mathbf{x}) + \rho$ where $\rho > 0$ is a tightening parameter that is to be chosen appropriately. Given that the constraint function is Lipschitz (via Assumption 3), this ensures that there is a "buffer zone" between the points that satisfy the tightened constraint $g(\mathbf{x}) + \rho \leq 0$ and the boundary of the true feasible set defined by $g(\mathbf{x}) \leq 0$. Therefore, by choosing $\rho$ proportional to the cost step-size $\eta$, we can ensure that the actions are feasible despite the adversarially-chosen cost

---

**Algorithm 1** OGD with Polyak Feasibility Steps

---
**input** initial action $\mathbf{x}_1 \in \mathbb{R}^d$, step size $\eta > 0$, tightening $\rho \geq 0$.
1: **for** $t = 1, 2, ..., T$ **do**
2:    Play $\mathbf{x}_t$ and receive $f_t$.
3:    Query constraint: $g_t = g(\mathbf{x}_t), \mathbf{s}_t \in \partial g(\mathbf{x}_t)$.
4:    Gradient descent: $\mathbf{y}_t = \mathbf{x}_t - \eta \nabla f_t(\mathbf{x}_t)$.
5:    Polyak feasibility step:[5]
   $\mathbf{x}_{t+1} = \Pi_{R\mathbb{B}} \left( \mathbf{y}_t - \frac{[g_t + \mathbf{s}_t^\top (\mathbf{y}_t - \mathbf{x}_t) + \rho]_+}{\|\mathbf{s}_t\|^2} \mathbf{s}_t \right)$.
6: **end for**

---

gradients. Note that tightening the constraint in this manner is a common technique in the constrained OCO literature, e.g. (Mahdavi et al., 2012; Jenatton et al., 2016).

**Polyak Feasibility Steps**  Using the ingredients discussed previously, we can put everything together to get our Polyak feasibility steps. Indeed, the step-size in line 5 is,

$$\frac{[g_t + \mathbf{s}_t^\top (\mathbf{y}_t - \mathbf{x}_t) + \rho]_+}{\|\mathbf{s}_t\|^2}, \quad (2)$$

where the numerator uses the first-order approximation in (1) and the tightening parameter $\rho$. As such (2) can be viewed as the Polyak step-size that uses a first-order approximation of the tightened constraint value $g(\mathbf{y}_t) + \rho$. In the analysis, we will see that this step-size ensures that the constraint value is greatly reduced in each step.

### 3.2. Guarantees

In this section, we give the regret guarantees and constraint satisfaction guarantees for Algorithm 1. In particular, the following theorem (Theorem 1) gives a bound on the regret and constraint violation for an arbitrary choice of algorithm parameters $\eta, \rho$. We will then show how these algorithm parameters can be chosen to get several different guarantees.

**Theorem 1.** *Let Assumptions 1, 2, 3 and 4 hold. Then, playing Algorithm 1 with $\eta > 0$ and $\rho \in [0, \epsilon]$ ensures that,*

$$\mathrm{Reg}_T \leq \frac{2R^2}{\eta} + \frac{\eta}{2}G_f^2 T + \frac{G_f \rho}{\sigma} T,$$

$$g(\mathbf{x}_t) \leq G_g \gamma^{(t-1)/2} \mathrm{dist}(\mathbf{x}_1, \mathcal{X}_\rho) + \frac{\eta G_g G_f}{1 - \sqrt{\gamma}} - \rho, \forall t$$

*where $\mathcal{X}_\rho = \{x \in \mathbb{R}^d : g(\mathbf{x}) \leq -\rho\}$ and $\gamma = 1 - \frac{\sigma^2}{G_g^2}$.*

In the following, Corollary 1 specifies a choice of $\eta$ and $\rho$ that ensures $\mathcal{O}(\sqrt{T})$ regret and anytime constraint satisfaction. Note that this result requires that the initial action $\mathbf{x}_1$ is strictly-feasible. We also point out that these guarantees

---

[5]When $\mathbf{s}_t = \mathbf{0}$, we take the update to be $\mathbf{x}_{t+1} = \Pi_{R\mathbb{B}} (\mathbf{y}_t)$.

still hold under a slightly weaker set of assumptions, which we discuss in Remark 2.

**Corollary 1.** *Let Assumptions 1, 2, 3 and 4 hold. Suppose that $g(\mathbf{x}_1) \leq -\alpha$ for some $\alpha \in (0, \epsilon]$, and let $\rho = \frac{\alpha}{\sqrt{T}}$ and $\eta = \frac{\xi\rho}{G_f G_g}$, where $\xi = 1 - \sqrt{\gamma}$. It follows that the actions chosen by Algorithm 1 satisfy,*

$$\text{Reg}_T \leq \left( \frac{2G_f G_g R^2}{\xi\alpha} + \frac{G_f \xi\alpha}{2G_g} + \frac{G_f \alpha}{\sigma} \right) \sqrt{T},$$

$$g(\mathbf{x}_t) \leq 0 \quad \forall t \in [T],$$

**Remark 2.** *The guarantees in Corollary 1 also hold under a slightly weaker set of assumptions. In particular, we only need that $\|\nabla f_t(\mathbf{x})\| \leq G_f$ for all $\mathbf{x} \in \mathcal{X}$ (rather than all $\mathbf{x} \in R\mathbb{B}$). We show this in Appendix C.4.*

Next, we give Corollary 2, which shows that, when $\mathbf{x}_1$ is in $R\mathbb{B}$ (and therefore is not necessarily feasible), then the algorithm can ensure anytime constraint satisfaction after $\mathcal{O}(\log(T))$ rounds. Corollary 2 also shows that when $T$ is sufficiently large, then the algorithm ensures that there is cumulative constraint satisfaction $\sum_{t=1}^{T} g(\mathbf{x}_t) \leq 0$. Note that the requirement that $T$ is sufficiently large is also used by prior work that shows cumulative constraint satisfaction (Mahdavi et al., 2012; Jenatton et al., 2016).

**Corollary 2.** *Let Assumptions 1, 2, 3 and 4 hold. Consider Algorithm 1 with $\mathbf{x}_1 \in R\mathbb{B}$, $\eta = \frac{\xi\epsilon}{2G_f G_g \sqrt{T}}$ and $\rho = \frac{\epsilon}{\sqrt{T}}$. Then, it holds that,*

$$\text{Reg}_T \leq \left( \frac{G_f G_g R^2}{\xi\epsilon} + \frac{G_f \xi\epsilon}{4G_g} + \frac{G_f \epsilon}{\sigma} \right) \sqrt{T},$$

$$g(\mathbf{x}_t) \leq 0 \quad \forall t \geq 1 + \frac{2G_g^2}{\sigma^2} \log \left( \frac{4G_g R \sqrt{T}}{\epsilon} \right).$$

*Furthermore, when $\sqrt{T} \geq \frac{4RG_g}{\epsilon\xi}$ it additionally holds that,*

$$\sum_{t=1}^{T} g(\mathbf{x}_t) \leq 0.$$

Lastly, we give another corollary below (Corollary 3) that allows a small amount of violation in each round, while eliminating the dependence on $G_g$ in the regret bound. In this case, the regret bound is $2RG_f \sqrt{T}$, which matches what is attained by online gradient descent using a full projection on to the action set in each round. At the same time, the constraint violation satisfies $g(x_t) = \mathcal{O}(\frac{1}{\sqrt{T}})$.

**Corollary 3.** *Let Assumptions 1, 2, 3 and 4 hold. Consider Algorithm 1 with $\mathbf{x}_1 \in R\mathbb{B}$, $\eta = \frac{2R}{G_f \sqrt{T}}$ and $\rho = 0$. Then, it holds that,*

$$\text{Reg}_T \leq 2RG_f \sqrt{T},$$

$$g(\mathbf{x}_t) \leq 2RG_g \exp \left( -\frac{\sigma^2(t-1)}{2G_g^2} \right) + \frac{2RG_g}{\xi\sqrt{T}} \quad \forall t \in [T]$$

### 3.3. Regret Analysis

In this section, we give the regret analysis. We separate the regret in to (I) the regret with respect to the tightened feasible set, and (II) the cost of tightening the feasible set,

$$\text{Reg}_T = \underbrace{\sum_{t=1}^{T} (f_t(\mathbf{x}_t) - f_t(\mathbf{x}_\rho^\star))}_{\text{Term I}} + \underbrace{\sum_{t=1}^{T} (f_t(\mathbf{x}_\rho^\star) - f_t(\mathbf{x}^\star))}_{\text{Term II}}, \tag{3}$$

where $\mathbf{x}_\rho^\star \in \arg\min_{\mathbf{x} \in \mathcal{X}_\rho} \sum_{t=1}^{T} f_t(\mathbf{x})$, and the tightened feasible set is,

$$\mathcal{X}_\rho := \{ \mathbf{x} \in \mathbb{R}^d : g(\mathbf{x}) \leq -\rho \}. \tag{4}$$

We start with Term I. The key observation is that the Polyak feasibility step (line 5) can be equivalently defined as the projection on to the halfspace $\mathcal{H}_t$ in (5) below (with the additional projection on to the ball). In fact, this halfspace $\mathcal{H}_t$ contains the tightened feasible set $\mathcal{X}_\rho$ and therefore projecting on to it will not increase the distance to any point in $\mathcal{X}_\rho$. This is stated precisely in Fact 1.

**Fact 1.** *The update for $\mathbf{x}_{t+1}$ in line 5 of Algorithm 1 is equivalent to $\mathbf{x}_{t+1} = \Pi_{R\mathbb{B}}(\Pi_{\mathcal{H}_t}(\mathbf{y}_t))$ where,*

$$\mathcal{H}_t = \{ \mathbf{x} \in \mathbb{R}^d : g_t + \mathbf{s}_t^\top (\mathbf{x} - \mathbf{x}_t) + \rho \leq 0 \}. \tag{5}$$

*Furthermore, it holds that $\mathcal{H}_t \supseteq \mathcal{X}_\rho$, and therefore that for all $\mathbf{x} \in \mathcal{X}_\rho$ and $\mathbf{v} \in \mathbb{R}^d$,*

$$\|\Pi_{\mathcal{H}_t}(\mathbf{v}) - \mathbf{x}\| \leq \|\mathbf{v} - \mathbf{x}\|. \tag{6}$$

In fact, the classical analysis of OGD (from Zinkevich (2003)) only requires that the projection does not increase the distance to the optimal action, which is ensured by (6) in Fact 1. Therefore, it follows from (6) and the standard analysis of OGD that we have a regret bound with respect to the tightened feasible set, and thus,

$$\text{Term I} \leq \frac{2R^2}{\eta} + \frac{\eta}{2} G_f^2 T. \tag{7}$$

We defer the complete proof of (7) to Appendix C.3.

Now, we look at Term II. We use Lemma 1 below, which provides a bound on the distance to $\mathcal{X}_\rho$ in terms of $g$. This lemma is conceptually similar to Theorem 7 in Mahdavi et al. (2012). However, it is more general in the sense that it is a general error bound on the constraint function, whereas Theorem 7 in Mahdavi et al. (2012) only provides a bound on the difference in costs between a point in the tightened feasible set and the original feasible set. The proof of Lemma 1 is given in Appendix C.2.

**Lemma 1.** *Let Assumptions 3 and 4 hold, and suppose that $\rho \in [0, \epsilon]$. Then, we have for all $\mathbf{x} \in R\mathbb{B}$ that,*

$$\text{dist}(\mathbf{x}, \mathcal{X}_\rho) \leq \frac{1}{\sigma} [g(\mathbf{x}) + \rho]_+ \tag{8}$$

It follows from Lemma 1 and the cost gradient bound (Assumption 2) that,

$$\text{Term II} = \sum_{t=1}^{T}(f_t(\mathbf{x}_\rho^\star) - f_t(\mathbf{x}^\star))$$

$$\leq \sum_{t=1}^{T}(f_t(\Pi_{\mathcal{X}_\rho}(\mathbf{x}^\star)) - f_t(\mathbf{x}^\star)) \quad \text{(a)}$$

$$\leq G_f T \text{dist}(\mathbf{x}^\star, \mathcal{X}_\rho) \quad \text{(b)}$$

$$\leq \frac{G_f T}{\sigma}[g(\mathbf{x}^\star) + \rho]_+ \leq \frac{G_f \rho}{\sigma}T, \quad \text{(c)}$$

where (a) uses the minimality of $\mathbf{x}_\rho^\star$, (b) uses the cost gradient bound, and (c) uses Lemma 1 and that $g(\mathbf{x}^\star) \leq 0$.

Combining the bounds on Term I and Term II yields,

$$\text{Reg}_T \leq \frac{2R^2}{\eta} + \frac{\eta}{2}G_f^2 T + \frac{G_f \rho}{\sigma}T,$$

which matches the bound in Theorem 1.

### 3.4. Feasibility Analysis

In this section, we give the feasibility analysis. The central result in this section is Lemma 2 below, which shows that the Polyak step-size shrinks the distance to a sub-level set of $g$. This lemma combines the classical analysis of the Polyak step-size from Polyak (1969) with Lemma 1.

**Lemma 2** (Polyak Step-size). *Let Assumption 3 and 4 hold, and suppose that $\rho \in [0, \epsilon]$. Furthermore, consider the $\rho$-sublevel set of $g$, $\mathcal{X}_\rho := \{\mathbf{x} \in \mathbb{R}^d : g(\mathbf{x}) + \rho \leq 0\}$. Consider any $\mathbf{x} \in R\mathbb{B}$ and $\mathbf{s} \in \partial g(\mathbf{x})$ such that $\mathbf{s} \neq \mathbf{0}$, and let,*

$$\mathbf{x}^+ = \Pi_{R\mathbb{B}}\left(\mathbf{x} - \frac{[g(\mathbf{x}) + \rho]_+}{\|\mathbf{s}\|^2}\mathbf{s}\right). \quad (9)$$

*Then, it holds that,*

$$\text{dist}^2(\mathbf{x}^+, \mathcal{X}_\rho) \leq \left(1 - \frac{\sigma^2}{G_g^2}\right)\text{dist}^2(\mathbf{x}, \mathcal{X}_\rho). \quad (10)$$

*Proof.* Let $\mathbf{v} = \Pi_{\mathcal{X}_\rho}(\mathbf{x})$ and denote $g_\rho(\mathbf{x}) := g(\mathbf{x}) + \rho$. First, note that if $g_\rho(\mathbf{x}) \leq 0$, then (9) becomes $\mathbf{x}^+ = \mathbf{x}$ and therefore $g_\rho(\mathbf{x}^+) = g_\rho(\mathbf{x}) \leq 0$ and (10) is satisfied with both sides 0. Therefore, we take $g_\rho(\mathbf{x}) > 0$ for the remainder (which ensures that $[g_\rho(\mathbf{x})]_+ = g_\rho(\mathbf{x})$). Then, it follows that,

$$\text{dist}^2(\mathbf{x}^+, \mathcal{X}_\rho) \leq \|\mathbf{x}^+ - \mathbf{v}\|^2$$

$$\leq \left\|\mathbf{x} - \frac{g_\rho(\mathbf{x})}{\|\mathbf{s}\|^2}\mathbf{s} - \mathbf{v}\right\|^2 \quad \text{(a)}$$

$$= \text{dist}^2(\mathbf{x}, \mathcal{X}_\rho) - 2\frac{g_\rho(\mathbf{x})}{\|\mathbf{s}\|^2}\mathbf{s}^\top(\mathbf{x} - \mathbf{v})$$

$$+ \frac{g_\rho(\mathbf{x})^2}{\|\mathbf{s}\|^2}$$

$$\leq \text{dist}^2(\mathbf{x}, \mathcal{X}_\rho) - \frac{g_\rho(\mathbf{x})^2}{\|\mathbf{s}\|^2} \quad \text{(b)}$$

$$\leq \text{dist}^2(\mathbf{x}, \mathcal{X}_\rho) - \frac{g_\rho(\mathbf{x})^2}{G_g^2} \quad \text{(c)}$$

$$\leq \left(1 - \frac{\sigma^2}{G_g^2}\right)\text{dist}^2(\mathbf{x}, \mathcal{X}_\rho), \quad \text{(d)}$$

where (a) uses the Pythagorean theorem of the projection, (b) uses that $\mathbf{s}^\top(\mathbf{x} - \mathbf{v}) \geq g_\rho(\mathbf{x}) - g_\rho(\mathbf{v}) \geq g_\rho(\mathbf{x})$ due to the fact that $\mathbf{s} \in \partial g(\mathbf{x})$ and $\mathbf{v} \in \mathcal{X}_\rho$, (c) uses that $\|\mathbf{s}\| \leq G_g$ due to Assumption 3, and (d) uses Lemma 1. $\square$

Using Lemma 2, we will show that the distance between the action $\mathbf{x}_t$ and the tightened feasible set $\mathcal{X}_\rho$ will not increase too much in each round, and therefore that $\mathbf{x}_t$ always stays close to $\mathcal{X}_\rho$. The key difficulty in doing so is that Lemma 2 uses the "exact" Polyak step-size, while Algorithm 1 uses the first-order approximation of the constraint value in the Polyak step-size (i.e. as in (2)). In order to handle this disparity, we introduce the "fictitious" iterate $\mathbf{z}_{t+1}$. We define $\mathbf{z}_{t+1}$ such that $\mathbf{z}_{t+1} = \mathbf{x}_t$ when $\mathbf{s}_t = \mathbf{0}$, and otherwise take,

$$\mathbf{z}_{t+1} = \Pi_{R\mathbb{B}}\left(\mathbf{x}_t - \frac{[g_t + \rho]_+}{\|\mathbf{s}_t\|^2}\mathbf{s}_t\right).$$

We can interpret $\mathbf{z}_{t+1}$ as a feasibility step that is taken directly from the previous action $\mathbf{x}_t$. Importantly, the update for $\mathbf{z}_{t+1}$ matches the form in (9) and therefore we will be able to directly apply Lemma 2 to analyze $\mathbf{z}_{t+1}$.

Using $\mathbf{z}_{t+1}$, we study the distance between $\mathbf{x}_{t+1}$ and $\mathcal{X}_\rho$,

$$\text{dist}(\mathbf{x}_{t+1}, \mathcal{X}_\rho) \leq \|\mathbf{x}_{t+1} - \Pi_{\mathcal{X}_\rho}(\mathbf{z}_{t+1})\|$$

$$\leq \underbrace{\|\mathbf{x}_{t+1} - \mathbf{z}_{t+1}\|}_{\text{Term I}} + \underbrace{\text{dist}(\mathbf{z}_{t+1}, \mathcal{X}_\rho)}_{\text{Term II}},$$

We start with Term I. First, note that $\mathbf{z}_{t+1}$ can be written as $\mathbf{z}_{t+1} = \Pi_{R\mathbb{B}}(\Pi_{\mathcal{H}_t}(\mathbf{x}_t))$, where $\mathcal{H}_t$ is the halfspace defined in (5). Therefore, it follows that,

$$\text{Term I} = \|\Pi_{R\mathbb{B}}(\Pi_{\mathcal{H}_t}(\mathbf{y}_t)) - \Pi_{R\mathbb{B}}(\Pi_{\mathcal{H}_t}(\mathbf{x}_t))\|$$

$$\leq \|\Pi_{\mathcal{H}_t}(\mathbf{y}_t) - \Pi_{\mathcal{H}_t}(\mathbf{x}_t)\|$$

$$\leq \|\mathbf{y}_t - \mathbf{x}_t\| \quad (11)$$

$$= \eta\|\nabla f_t(\mathbf{x}_t)\|$$

$$\leq \eta G_f,$$

where the first two lines use the non-expansiveness of the projection, and the last line uses the cost gradient bound.

Next, we look at Term II. If $\mathbf{s}_t = \mathbf{0}$, then it holds that $\mathbf{z}_{t+1} = \mathbf{x}_t$ and $g(\mathbf{x}_t) = \min_{\mathbf{x}} g(\mathbf{x}) \leq -\rho$, which implies that Term II = 0. Alternatively, if $\mathbf{s}_t \neq \mathbf{0}$, then applying Lemma 2 yields $(\text{Term II})^2 \leq \gamma \text{dist}^2(\mathbf{x}_t, \mathcal{X}_\rho)$, where $\gamma := 1 - \frac{\sigma^2}{G_g^2}$. Therefore, it holds in any case that,

$$\text{Term II} \leq \sqrt{\gamma}\text{dist}(\mathbf{x}_t, \mathcal{X}_\rho)$$

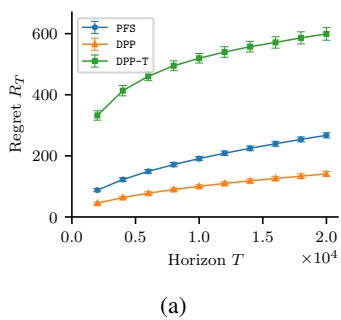 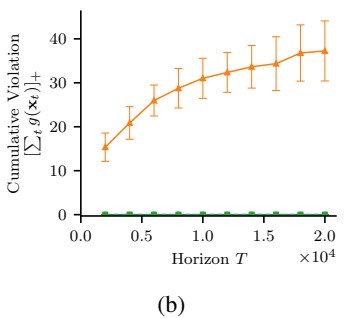 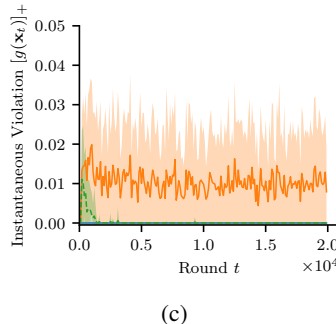

| (a) | (b) | (c) |

*Figure 1.* Experiment results for our algorithm (labeled PFS), alongside the algorithm from Yu et al. (2017) (labeled DPP). We also include a version of the algorithm in Yu et al. (2017) where the constraint is tightened (labeled DPP-T). The points indicate the average over 30 trials and the error bars and shading are ±1 standard deviation.

Combining Term I and Term II yields,

$$\text{dist}(\mathbf{x}_{t+1}, \mathcal{X}_\rho) \leq \sqrt{\gamma}\text{dist}(\mathbf{x}_t, \mathcal{X}_\rho) + \eta G_f$$

$$\leq \gamma^{t/2}\text{dist}(\mathbf{x}_1, \mathcal{X}_\rho) + \eta G_f \sum_{s=0}^{t-1}(\sqrt{\gamma})^s$$

$$\leq \gamma^{t/2}\text{dist}(\mathbf{x}_1, \mathcal{X}_\rho) + \frac{\eta G_f}{1 - \sqrt{\gamma}}$$

where we apply the bound recursively in the second line, and use the fact that $\gamma \in [0, 1)$ in the third line. Then, applying the the subgradient bound (Assumption 3),

$$g(\mathbf{x}_t) = g(\mathbf{x}_t) - g(\Pi_{\mathcal{X}_\rho}(\mathbf{x}_t)) + g(\Pi_{\mathcal{X}_\rho}(\mathbf{x}_t))$$

$$\leq G_g\text{dist}(\mathbf{x}_t, \mathcal{X}_\rho) - \rho$$

$$\leq G_g\gamma^{(t-1)/2}\text{dist}(\mathbf{x}_1, \mathcal{X}_\rho) + \frac{\eta G_g G_f}{1 - \sqrt{\gamma}} - \rho.$$

This matches the constraint violation guarantee in Theorem 1.

## 4. Numerical Experiments

Although our primary contribution is our theoretical results, we also give numerical experiments to demonstrate the functionality of the algorithm and provide some empirical verification of the theoretical results.[6] In these experiments, we benchmark the performance of our algorithm with the algorithm from Yu et al. (2017) as it has the best regret bound among those in Table 1.

We consider a 2-dimensional toy setting with quadratic cost functions $f_t(\mathbf{x}) = 3\|\mathbf{x} - \mathbf{v}_t\|^2$ and linear constraints $A\mathbf{x} \leq \mathbf{b}$. We generate $\mathbf{v}_t$ by sampling uniformly from $[0, 1]^2$, and take $A = [I - I]^\top$ and $\mathbf{b} = 0.5\mathbf{1}$, where we use $I$ to denote the $2 \times 2$ identity matrix. Therefore, we can define the constraint function as $g(\mathbf{x}) = \max_{i \in [4]} \mathbf{a}_i^\top \mathbf{x} - b_i$, where $\mathbf{a}_i$

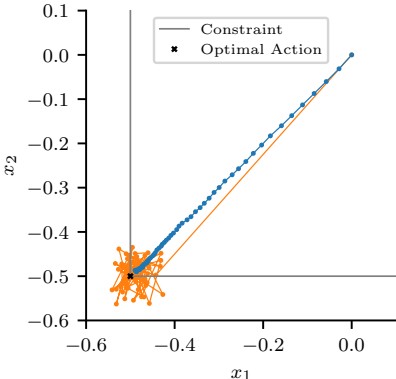

*Figure 2.* Actions chosen by our algorithm and the one from Yu et al. (2017) in an experiment.

and $b_i$ are the $i$th row of $A$ and $\mathbf{b}$ respectively.[7] Accordingly, we take $G_f = \sqrt{2}$, $R = 1$, $G_g = 1$, $\epsilon = 0.25$ and $\sigma = \frac{1}{\sqrt{2}}$.

In this setting, we implement Algorithm 1 (labeled PFS) with the algorithm parameters chosen according to Theorem 1, as well as the algorithm in Yu et al. (2017) (labeled DPP) with the algorithm parameters chosen according to their Theorem 1, i.e. $\alpha = T$, $V = \sqrt{T}$. We also implement the algorithm from Yu et al. (2017) with a tightened constraint $g_\rho(\mathbf{x}) := g(\mathbf{x}) + \rho$ for $\rho \in [0, \epsilon]$, as is used in our algorithm and that in Mahdavi et al. (2012) and Jenatton et al. (2016). Same as the aforementioned algorithms, we use a decreasing $\rho = \min(\epsilon, \frac{c}{\sqrt{T}})$, where $c > 0$ is a parameter that we tune to reduce the violation. We choose $c = 20$ to ensure that there is a small amount of constraint violation. The tightened version of DPP is labeled DPP-T.

The results are shown in Figure 1. Precisely, Figure 1a and Figure 1b show the regret and cumulative violation for $T = 2\times10^3, 4\times10^3, ..., 2\times10^4$, where the marker indicates

---

[6]Our experiment code is available at https://github.com/shutch1/OCO-Polyak-Feasibility-Steps.

[7]Note that the constraint can also be written with the infinity-norm: $g(\mathbf{x}) = \|x\|_\infty - b$.

the mean over the 30 trials and the errorbar indicates the standard deviation. Figure 1c shows the instantaneous violation $[g(\mathbf{x}_t)]_+$ at each round for a fixed $T = 2 \times 10^4$, where the line shows the mean over 30 trials and the shading indicates the standard deviation. In these results, DPP enjoys smaller regret than PFS. However, DPP incurs constraint violation, while PFS does not incur constraint violation. By augmenting DPP with a tightened constraint, the constraint violation can be substantially reduced as shown for DPP-T. However, this also results in a larger regret.

To provide intuition on the operation of our algorithm, we also include Figure 2, which shows every 300th action chosen by the algorithms in one simulation trial. In this plot, it can be seen that PFS takes a conservative approach in gradually approaching the constraint boundary, which ensures constraint satisfaction at the cost of larger regret.

## 5. Conclusion

In this work, we give an algorithm for constrained OCO that uses Polyak feasibility steps to ensure anytime constraint satisfaction $g(\mathbf{x}_t) \le 0 \; \forall t$ and $\mathcal{O}(\sqrt{T})$ regret, while only receiving feedback on the function value and subgradient at the played actions. We foresee this approach being particularly relevant to safety-critical applications, where constraints must be satisfied despite having only limited constraint information.

## Acknowledgements

This work was supported by the National Science Foundation under grant #2330154.

## Impact Statement

This paper presents work whose goal is to advance the field of Machine Learning. There are many potential societal consequences of our work, none which we feel must be specifically highlighted here.

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

## A. Extension of Prior Work to No Cumulative Constraint Violation

In this section, we show how the prior work Yu et al. (2017) and Yuan & Lamperski (2018) can be extended to show cumulative constraint satisfaction $\sum_{t=1}^{T} g(\mathbf{x}_t) \leq 0$ under Assumption 4 (which is the same as Assumption 1 in Mahdavi et al. (2012)). This follows a similar approach to Theorem 8 in (Mahdavi et al., 2012), but we give it for completeness.

The algorithm in Yu et al. (2017) guarantees $\text{Reg}_T \leq C_R \sqrt{T}$ and $\sum_{t=1}^{T} g(\mathbf{x}_t) \leq C_V \sqrt{T}$ for some constants $C_R, C_V > 0$. Therefore, by applying this algorithm to the tightened constraint $g_\rho(\mathbf{x}) := g(\mathbf{x}) + \rho$, we can guarantee that $\sum_{t=1}^{T} g_\rho(\mathbf{x}_t) \leq C_V \sqrt{T}$ and,

$$\text{Reg}_T^\rho := \sum_{t=1}^{T} f_t(\mathbf{x}_t) - \min_{\mathbf{x} \in \mathcal{X}_\rho} \sum_{t=1}^{T} f_t(\mathbf{x}) \leq C_R \sqrt{T}, \tag{12}$$

where $\mathcal{X}_\rho$ is the sub-level set of the tightened constraint defined in (4). Choosing $\rho = \min\left(\epsilon, \frac{C_V}{\sqrt{T}}\right)$ (where $\epsilon$ is defined in Assumption 4) and taking $T \geq \frac{C_V^2}{\epsilon^2}$ ensures that,

$$\sum_{t=1}^{T} g(\mathbf{x}_t) = \sum_{t=1}^{T} g_\rho(\mathbf{x}_t) - \rho T \leq C_V \sqrt{T} - \min\left(\epsilon, \frac{C_V}{\sqrt{T}}\right) T = 0. \tag{13}$$

Then, with $\mathbf{x}_\rho^\star \in \arg\min_{\mathbf{x} \in \mathcal{X}_\rho} \sum_{t=1}^{T} f_t(\mathbf{x})$,

$$\text{Reg}_T = \text{Reg}_T^\rho + \sum_{t=1}^{T} (f_t(\mathbf{x}_\rho^\star) - f_t(\mathbf{x}^\star)) \leq \text{Reg}_T^\rho + \sum_{t=1}^{T} (f_t(\Pi_{\mathcal{X}_\rho}(\mathbf{x}^\star)) - f_t(\mathbf{x}^\star)) \leq \text{Reg}_T^\rho + TG_g\text{dist}(\mathbf{x}^\star, \mathcal{X}_\rho)$$

Then, applying Lemma 1 and (12), and using the fact that $g(\mathbf{x}^\star) \leq 0$ yields,

$$\text{Reg}_T \leq C_R \sqrt{T} + T\frac{G_g}{\sigma}[g(\mathbf{x}^\star) + \rho]_+ \leq C_R \sqrt{T} + T\frac{G_g}{\sigma}\rho \leq C_R \sqrt{T} + \frac{G_g C_V}{\sigma}\sqrt{T}.$$

Therefore, we have shown that the algorithm from Yu et al. (2017) can be extended to show $\mathcal{O}(\sqrt{T})$ regret and cumulative constraint satisfaction $\sum_{t=1}^{T} g(\mathbf{x}_t) \leq 0$, provided that Assumption 4 holds and that $T$ is sufficiently large, i.e. $T \geq \frac{C_V^2}{\epsilon^2}$. Note that the requirement that $T$ is sufficiently large is also required by other works that guarantee cumulative constraint satisfaction, e.g. (Mahdavi et al., 2012; Jenatton et al., 2016).

We can use a similar process to give guarantees for Yuan & Lamperski (2018). The original guarantees in Yuan & Lamperski (2018) are of the form $\text{Reg}_T \leq C_R T^{\max(\beta, 1-\beta)}$ and $\sum_{t=1}^{T} [g(\mathbf{x}_t)]_+ \leq C_V T^{1-\beta/2}$. Therefore, choosing $\rho = \min\left(\epsilon, \frac{C_V}{T^{\beta/2}}\right)$ and taking $T \geq \frac{C_V^{2/\beta}}{\epsilon^{2/\beta}}$ ensures that $\sum_{t=1}^{T} g(\mathbf{x}_t) \leq 0$. Also, the resulting regret is,

$$\text{Reg}_T \leq C_R T^{\max(\beta, 1-\beta)} + \frac{G_g C_V}{\sigma}T^{1-\beta/2}.$$

The order of the bound $\max(\beta, 1 - \beta, 1 - \beta/2)$ is minimized when $\beta = \frac{2}{3}$ and therefore, we get that $\text{Reg}_T = \mathcal{O}(T^{2/3})$. Note that Yuan & Lamperski (2018) also gives a bound of the form $\sum_{t=1}^{T}([g(\mathbf{x}_t)]_+)^2 \leq C_V T^{1-\beta}$, and therefore one might hope to guarantee $\sum_{t=1}^{T}([g(\mathbf{x}_t)]_+)^2 \leq 0$. Unfortunately, the analysis approach does not immediately show this because the tightening parameter $\rho$ cannot be used to cancel out the violation as in (13).

## B. Comparison of Algorithm with Related Work

In this section, we discuss the differences between our algorithm design and the algorithm design in Zhang et al. (2025), which studies constrained variational inequalities. Although Zhang et al. (2025) studies a different setting and therefore uses a different analysis, there are some apparent similarities in algorithm design that we discuss here. To illustrate the similarities and differences, we can write our algorithm in a different form that allows for more direct comparison. In particular, when the cost functions are fixed $f_t = f$ and the constraint function is smooth, we can write *our algorithm* (with $\rho = 0$) as,

$$\mathbf{x}_{t+1} = \Pi_{R\mathbb{B}}\left(\mathbf{x}_t - \eta\nabla f(\mathbf{x}_t) - \frac{[g(\mathbf{x}_t) - \eta\nabla f(\mathbf{x}_t)^\top \nabla g(\mathbf{x}_t)]_+}{||\nabla g(\mathbf{x}_t)||^2}\nabla g(\mathbf{x}_t)\right), \tag{14}$$

and the algorithm in Zhang et al. (2025) can be written as (see their Algorithm 2),

$$\mathbf{x}_{t+1} = \mathbf{x}_t - \eta \nabla f(\mathbf{x}_t) - \eta \frac{[\alpha g(\mathbf{x}_t) - \nabla f(\mathbf{x}_t)^\top \nabla g(\mathbf{x}_t)]_+}{\|\nabla g(\mathbf{x}_t)\|^2} \nabla g(\mathbf{x}_t). \tag{15}$$

In both algorithms, the step-size $\eta$ is $\Theta(\frac{1}{\sqrt{T}})$ and $\alpha = G_g/R$ in (15). As such, we can see several differences:

- Zhang et al. (2025) applies the step-size $\eta$ to the entire third term in (15), while our algorithm in (14) applies the step size only to $\nabla f(\mathbf{x}_t)^\top \nabla g(\mathbf{x}_t)$ within the third term.

- Zhang et al. (2025) applies the scaling $\alpha$ to $g(\mathbf{x}_t)$ in (15), while our algorithm uses no such scaling in (14).

- Zhang et al. (2025) uses an additional constraint to ensure that the iterates are bounded, whereas we use a projection on to $R\mathbb{B}$.

## C. Missing proofs

### C.1. Proof of Remark 1

In this section, we show that Slater's condition implies Assumption 4, provided that Assumption 1 holds.

**Proposition 1.** *Let Assumption 1 hold. Then, suppose that Slater's condition holds, i.e. there exists positive real $\xi$ and $\mathbf{y} \in \mathbb{R}^d$ such that $g(\mathbf{y}) \leq -\xi$. It follows that Assumption 4 holds with $\epsilon = c\xi$ and $\sigma = (1-c)\frac{\xi}{2R}$ for any $c \in (0,1)$.*

*Proof.* It holds that $\mathcal{X}' = \{\mathbf{x} \in \mathbb{R}^d : g(\mathbf{x}) = -\epsilon\}$ is non-empty because there exists $\mathbf{z}, \mathbf{y}$ such that $g(\mathbf{z}) = 0$ (given that $\mathcal{X}$ is nonempty and compact) and $g(\mathbf{y}) \leq -\xi$. Specifically, since $-\epsilon \in [-\xi, 0]$ and $g$ is continuous (via Corollary 10.1.1 in (Rockafellar, 1970)), it holds that there exists $\mathbf{z}$ such that $g(\mathbf{z}) = -\epsilon$.

Then, we show that $\|\partial g(\mathbf{x})\| \geq \sigma$ for all $\mathbf{x} \in \mathcal{X}'$. Indeed for all $\mathbf{x} \in \mathcal{X}'$, it holds for all $\mathbf{s} \in \partial g(\mathbf{x})$ that,

$$- \xi \geq g(\mathbf{y}) \geq g(\mathbf{x}) + \mathbf{s}^\top(\mathbf{y} - \mathbf{x}) = -\epsilon + \mathbf{s}^\top(\mathbf{y} - \mathbf{x}) \geq -\epsilon - \|\mathbf{s}\|\|\mathbf{y} - \mathbf{x}\| \geq -\epsilon - 2\|\mathbf{s}\|R$$

$$\implies \quad \|\mathbf{s}\| \geq \frac{\xi - \epsilon}{2R} = (1-c)\frac{\xi}{2R} = \sigma,$$

where we use the fact that $\|\mathbf{y} - \mathbf{x}\| \leq 2R$ given that $\mathbf{x}, \mathbf{y} \in \mathcal{X} \subseteq R\mathbb{B}$. $\qquad \square$

### C.2. Proof of Lemma 1

In this section, we give the proof of Lemma 1. This proof relies on the following lemma, which gives several properties of the projection on to the sub-level set of a convex function.

**Lemma 3.** *Consider a closed convex function $h : \mathbb{R}^d \to \mathbb{R}$, where we use the notation $\mathcal{S} = \{x \in \mathbb{R}^d : h(x) \leq 0\}$. Let $\mathcal{K}$ be a convex set such that $\mathcal{S} \subseteq \mathcal{K}$. Assume that, for all $\mathbf{x} \in \mathcal{K}$, it holds that $\|\mathbf{s}\| \leq G$ for all $\mathbf{s} \in \partial h(\mathbf{x})$, and when $h(\mathbf{x}) = 0$, it holds that $\|\mathbf{s}\| \geq \sigma$ for all $s \in \partial h(\mathbf{x})$. Then, take any $\mathbf{x} \in \mathcal{K}$ such that $h(\mathbf{x}) > 0$. Let $\mathbf{z} = \Pi_{\mathbf{x}:h(\mathbf{x})\leq 0}(\mathbf{x})$ and $\mathbf{s}_x \in \partial h(\mathbf{x})$. There exists $\gamma \geq 0$ and $\mathbf{s}_z \in \partial h(\mathbf{z})$ such that:*

1. $h(\mathbf{z}) = 0$,

2. $\mathbf{x} - \mathbf{z} = \gamma \frac{\mathbf{s}_z}{\|\mathbf{s}_z\|}$,

3. $\gamma \leq \frac{h(\mathbf{x})}{\|\mathbf{s}_z\|}$,

4. $\|\mathbf{s}_z\|^2 \leq \mathbf{s}_x^\top \mathbf{s}_z$,

5. $\|\mathbf{s}_x\| \geq \|\mathbf{s}_z\|$.

*Proof.* First, note that the $\mathcal{S} := \{\mathbf{x} \in \mathbb{R}^d : h(\mathbf{x}) \leq 0\}$ is a closed and convex set so this projection is well-defined. Then, we show each point in the following.

**1.** Suppose the contrary, i.e. that $h(\mathbf{z}) \leq -\epsilon$ for some $\epsilon > 0$. It follows that $\mathbf{z} + \frac{\epsilon}{G}\mathbb{B} \subseteq \mathcal{S}$ as

$$\sup_{\mathbf{v} \in \mathbb{B}} h(z + \frac{\epsilon}{G}\mathbf{v}) \leq h(\mathbf{z}) + G\frac{\epsilon}{G} \leq -\epsilon + \epsilon = 0.$$

Therefore, $\mathbf{z}' = \mathbf{z} + \frac{\mathbf{x}-\mathbf{z}}{\|\mathbf{x}-\mathbf{z}\|}\frac{\epsilon}{G} \in \mathcal{S}$ and

$$\|\mathbf{z}' - \mathbf{x}\| = \left\|\mathbf{z} + \frac{\mathbf{x}-\mathbf{z}}{\|\mathbf{x}-\mathbf{z}\|}\frac{\epsilon}{G} - \mathbf{x}\right\| = \left(1 - \frac{\epsilon}{\|\mathbf{x}-\mathbf{z}\|G}\right)\|\mathbf{z}-\mathbf{x}\| < \|\mathbf{z}-\mathbf{x}\|,$$

contradicting the optimality of $\mathbf{z}$ as the minimizer of the distance to $\mathcal{S}$.

**2.** Since $\mathbf{z} = \Pi_{\mathcal{S}}(\mathbf{x})$ and $\mathcal{S}$ is a closed convex set, we know that the vector $\mathbf{x} - \mathbf{z}$ is normal to $\mathcal{S}$ at $\mathbf{z}$. Furthermore, $\mathcal{S} = \{\mathbf{x} : h(\mathbf{x}) \leq h(\mathbf{z}) = 0\}$ and the conditions of the lemma ensures that $\mathbf{0}$ is not a subgradient of $h$ at $\mathbf{z}$ (and therefore $\mathbf{z}$ is not a minimum of $h$) so (Rockafellar, 1970) (Corollary 23.7.1) tells us that there exists $\lambda \geq 0$ such that $\mathbf{x} - \mathbf{z} \in \lambda \partial h(\mathbf{z})$. Therefore, there exists $\gamma \geq 0$ and $\mathbf{s}_z \in \partial h(\mathbf{z})$ such that $\mathbf{x} - \mathbf{z} = \gamma \frac{\mathbf{s}_z}{\|\mathbf{s}_z\|}$.

**3.** It follows from the definition of subgradient of $h$ that,

$$h(\mathbf{x}) \geq h(\mathbf{z}) + \mathbf{s}_z^\top (\mathbf{x} - \mathbf{z}) = \mathbf{s}_z^\top (\mathbf{x} - \mathbf{z}) = \gamma \frac{\mathbf{s}_z^\top \mathbf{s}_z}{\|\mathbf{s}_z\|} = \gamma\|\mathbf{s}_z\|.$$

Rearranging yields,

$$\gamma \leq \frac{h(\mathbf{x})}{\|\mathbf{s}_z\|}.$$

**4.** From the monotonicity of the subgradients of a convex function, it holds that

$$0 \leq (\mathbf{s}_z - \mathbf{s}_x)^\top (\mathbf{z} - \mathbf{x}) = -\frac{\gamma}{\|\mathbf{s}_z\|}(\mathbf{s}_z - \mathbf{s}_x)^\top \mathbf{s}_z$$

$$\implies \quad 0 \geq (\mathbf{s}_z - \mathbf{s}_x)^\top \mathbf{s}_z$$

$$\implies \quad \|\mathbf{s}_z\|^2 \leq \mathbf{s}_x^\top \mathbf{s}_z.$$

**5.** From #4 and Cauchy-Schwarz,

$$\|\mathbf{s}_z\|^2 \leq \mathbf{s}_x^\top \mathbf{s}_z \leq \|\mathbf{s}_x\|\|\mathbf{s}_z\| \quad \implies \quad \|\mathbf{s}_x\| \geq \|\mathbf{s}_z\|.$$

$\square$

Then, we give the proof of Lemma 1 in the following.

*Proof of Lemma 1.* We use the notation $g_\rho(x) := g(x) + \rho$. First note that when $g_\rho(\mathbf{x}) \leq 0$, then $\mathbf{x} \in \mathcal{X}_\rho$ and therefore (8) holds with both sides zero. Next, we consider the case where $g_\rho(\mathbf{x}) > 0$. To do so, we will first show that the subgradient norm of $g_\rho(\mathbf{x})$ is lower bounded at all $\mathbf{x}$ such that $g_\rho(\mathbf{x}) = 0$. In particular, consider any $\mathbf{x}$ such that $g_\rho(\mathbf{x}) = 0$ (which necessarily satisfies $\mathbf{x} \in R\mathbb{B}$), and let $\mathbf{z} = \Pi_{g(\mathbf{x}) \leq -\epsilon}(\mathbf{x})$ and $\mathbf{s}_z \in \partial g(\mathbf{z})$. Then, it follows from Assumption 4 that $\|\mathbf{s}_z\| \geq \sigma$ and therefore we can apply Lemma 3 #5 by setting $\mathcal{K} \leftarrow R\mathbb{B}$, $h \leftarrow g + \epsilon$, $\sigma \leftarrow \sigma$ to get that for all $\mathbf{s}_x \in \partial g(\mathbf{x})$,

$$\|\mathbf{s}_x\| \geq \|\mathbf{s}_z\| \geq \sigma.$$

Note that we have used the fact that $\partial(g(\mathbf{x}) + \epsilon) = \partial g_\rho(\mathbf{x}) = \partial g(\mathbf{x})$ since they only vary by a constant. Since we know that the subgradient norm of $g_\rho$ is lower bounded at the boundary of $\mathcal{X}_\rho$, we can apply Lemma 3 #3 with $\mathcal{K} \leftarrow R\mathbb{B}$, $h \leftarrow g_\rho$, $G \leftarrow G_g$ and $\sigma \leftarrow \sigma$ to get that,

$$\text{dist}(\mathbf{x}, \mathcal{X}_\rho) \leq \frac{g_\rho(\mathbf{x})}{\|\mathbf{s}_z\|} \leq \frac{g_\rho(\mathbf{x})}{\sigma},$$

which verifies (8). $\square$

## C.3. Proof of (7)

In this section, we prove (7). This combines Fact 1 with the classical online gradient descent analysis from (Zinkevich, 2003).

*Proof of* (7). Given that $\mathbf{x}_\rho^\star \in \mathcal{X}_\rho \subseteq \mathcal{H}_t$ and $\mathbf{x}_\rho^\star \in \mathcal{X}_\rho \subseteq \mathcal{X} \subseteq R\mathbb{B}$ (and both $\mathcal{H}_t$ and $R\mathbb{B}$ are convex), it holds that,

$$
\begin{aligned}
\|\mathbf{x}_{t+1} - \mathbf{x}_\rho^\star\|^2 &= \|\Pi_{R\mathbb{B}}(\Pi_{\mathcal{H}_t}(\mathbf{y}_{t+1})) - \mathbf{x}_\rho^\star\|^2 \\
&\leq \|\Pi_{\mathcal{H}_t}(\mathbf{y}_{t+1}) - \mathbf{x}_\rho^\star\|^2 \\
&\leq \|\mathbf{y}_{t+1} - \mathbf{x}_\rho^\star\|^2 \\
&\leq \|\mathbf{x}_t - \eta \nabla f_t(\mathbf{x}_t) - \mathbf{x}_\rho^\star\|^2 \\
&= \|\mathbf{x}_t - \mathbf{x}_\rho^\star\|^2 - 2\eta \nabla f_t(\mathbf{x}_t)^\top (\mathbf{x}_t - \mathbf{x}_\rho^\star) + \eta^2 \|\nabla f_t(\mathbf{x}_t)\|^2.
\end{aligned}
$$

Using this and the convexity of $f_t$, it follows that

$$
\begin{aligned}
\text{Term I} &= \sum_{t=1}^{T} (f_t(\mathbf{x}_t) - f_t(\mathbf{x}_\rho^\star)) \\
&\leq \sum_{t=1}^{T} \nabla f_t(\mathbf{x}_t)^\top (\mathbf{x}_t - \mathbf{x}_\rho^\star) \\
&\leq \frac{1}{2\eta} \sum_{t=1}^{T} (\|\mathbf{x}_t - \mathbf{x}_\rho^\star\|^2 - \|\mathbf{x}_{t+1} - \mathbf{x}_\rho^\star\|^2) + \frac{\eta}{2} \sum_{t=1}^{T} \|\nabla f_t(\mathbf{x}_t)\|^2 \\
&= \frac{1}{2\eta} (\|\mathbf{x}_1 - \mathbf{x}_\rho^\star\|^2 - \|\mathbf{x}_{T+1} - \mathbf{x}_\rho^\star\|^2) + \frac{\eta}{2} \sum_{t=1}^{T} \|\nabla f_t(\mathbf{x}_t)\|^2 \\
&\leq \frac{2}{\eta} R^2 + \frac{\eta}{2} G_f^2 T
\end{aligned}
$$

where the third inequality uses Assumptions 1 and 2. $\qquad \square$

## C.4. Proof of Remark 2

In this section, we show that the guarantees of Corollary 1 hold under slightly weaker assumptions. In particular, we use the following Assumption 5 instead of Assumption 2. Assumption 5 is weaker than Assumption 2 in that it only requires that the cost gradients are bounded for all $\mathbf{x} \in \mathcal{X}$, rather than all $\mathbf{x} \in R\mathbb{B}$.

**Assumption 5.** *There exists positive real $G_f$ such that $\|\nabla f_t(\mathbf{x})\| \leq G_f$ for all $\mathbf{x} \in \mathcal{X}$ and $t \in [T]$.*

We then state the guarantees as a proposition and prove it.

**Proposition 2.** *Let Assumptions 1, 3, 4 and 5 hold. Suppose that $g(\mathbf{x}_1) \leq -\alpha$ for some $\alpha \in (0, \epsilon]$, and let $\rho = \frac{\alpha}{\sqrt{T}}$ and $\eta = \frac{\xi\rho}{G_f G_g}$, where $\xi = 1 - \sqrt{\gamma}$. It follows that the actions chosen by Algorithm 1 satisfy,*

$$
\begin{aligned}
\text{Reg}_T &\leq \left( \frac{2G_f G_g R^2}{\xi\alpha} + \frac{G_f \xi\alpha}{2G_g} + \frac{G_f \alpha}{\sigma} \right) \sqrt{T}, \\
g(\mathbf{x}_t) &\leq 0 \quad \forall t \in [T],
\end{aligned}
$$

*Proof.* First, we show that $\|\nabla f_t(\mathbf{x}_t)\| \leq G_f$ for all $t \in [T]$ by induction over the first $\tau$ rounds. The base case holds because $\mathbf{x}_1 \in \mathcal{X}_\rho \subseteq \mathcal{X}$ and therefore $\|\nabla f_t(\mathbf{x}_t)\| \leq G_f$ by Assumption 5. For the induction step, suppose that $\|\nabla f_t(\mathbf{x}_t)\| \leq G_f$ for all $t \in [\tau]$. Then, note that the feasibility analysis in Section 3.4 only requires that the cost gradient is bounded at the previously played actions. In particular, this bound is applied in (11). Therefore, it holds that,

$$
g(\mathbf{x}_{\tau+1}) \leq G_g \gamma^{t/2} \text{dist}(\mathbf{x}_1, \mathcal{X}_\rho) + \frac{\eta G_g G_f}{1 - \sqrt{\gamma}} - \rho,
$$

implying that $g(\mathbf{x}_{\tau+1}) \leq 0$ with the specified choice of algorithm parameters. Since $\mathbf{x}_{\tau+1} \in \mathcal{X}$, it holds that $\|\nabla f_{\tau+1}(\mathbf{x}_{\tau+1})\| \leq G_f$, and together with the induction hypothesis, it holds that $\|\nabla f_t(\mathbf{x}_t)\| \leq G_f$ for all $t \in [\tau + 1]$ and the induction step is complete. Thus, it holds that $\|\nabla f_t(\mathbf{x}_t)\| \leq G_f$ for all $t \in [T]$. Along the way, we have also shown that $\mathbf{x}_t \in \mathcal{X}$ for all $t \in [T]$.

Next, we show that, because the cost gradients are bounded at the actions, the regret bound holds. In particular, the analysis of Term I in the regret analysis (see Section 3.3) holds because it only requires that $\|\nabla f_t(\mathbf{x}_t)\| \leq G_f$ for all $t \in [T]$ (as shown in Section C.3). The analysis of Term II requires that $f_t(\Pi_{\mathcal{X}_\rho}(\mathbf{x}^\star)) - f_t(\mathbf{x}^\star) \leq G_f \|\Pi_{\mathcal{X}_\rho}(\mathbf{x}^\star) - \mathbf{x}^\star\|$. This holds because the cost gradients are assumed to be bounded on $\mathcal{X}$ and therefore all $f_t$ are Lipschitz on $\mathcal{X}$. $\qquad\square$

