# OpenReview forum: "Constrained Online Convex Optimization with Polyak Feasibility Steps"
_ICML.cc/2025/Conference — ICML 2025 poster_

### Official Review · Reviewer_JVnc · 2025-03-05

**Overall Recommendation:** 3

**Summary:**

The paper investigates online convex optimization subject to fixed convex constraint by incorporating Polyak feasibility steps. This leads to a sublinear regret and different constraint violation guarantees: no violation if initial start satisfies constraint, no violation after $O(\log T)$ rounds or cumulatively constrained evaluation is small.

**Claims And Evidence:**

The results support the claim

**Essential References Not Discussed:**

Well referenced.

**Experimental Designs Or Analyses:**

The experiment compared an extension of previous work with current work. Theoretically they can achieve the same bound, empirically the current work seems to be an interpretation between low regret and low violation cost.

**Methods And Evaluation Criteria:**

The paper use standard metric, regret and cost associated with constraint violation

**Other Comments Or Suggestions:**

typo: line 316 : y_{t} instead of y_{t+1}

section 3.4 begins with using eqn (9), and quoted Fact 1 which was with reference of eqn (2) or line 5 of Algorithm 1, $\rpo$ in different place means different things, although it does not affect the result, but in between line 382 left - line 331 right causes unnecessary confusion.

Result reporting: As theorem 1 is already a consequence of plugging in hyper parameters thus it does not allow seeing how parameter changes yield to corollaries. It only became clear how hyper parameters change after reading section 3.3 and 3.4

**Other Strengths And Weaknesses:**

The paper is well written and easy to follow. The explanation of incorporating Polyak feasibility steps is sensible and theoretically supported.

**Questions For Authors:**

the regret scales with multiplicative Lipschitz constants $G_f G_g$, from current analysis it does not seem to be possible to have additative dependence $G_f + G_g$, is this common?

**Relation To Broader Scientific Literature:**

An new approach to solve constrained OCO problem with comparatively the same rate as the extension from previous work. (The extension was done by the authors)

**Theoretical Claims:**

All steps were checked.

---

> ### Author Rebuttal · Authors · 2025-04-01
>
> We appreciate your detailed review and positive appraisal of the work. We discuss each of your points below.
>
> **Q.1. typo: line 316 : $y_{t}$ instead of $y_{t+1}$**
>
> You are correct. Thank you for pointing this out.
>
> **Q.2. section 3.4 begins with using eqn (9), and quoted Fact 1 which was with reference of eqn (2) or line 5 of Algorithm 1, $\rho$ in different place means different things, although it does not affect the result, but in between line 382 left - line 331 right causes unnecessary confusion.**
>
> Thank you for pointing this out. We will adjust our notation to make it clearer.
>
> **Q.3. Result reporting: As theorem 1 is already a consequence of plugging in hyper parameters thus it does not allow seeing how parameter changes yield to corollaries. It only became clear how hyper parameters change after reading section 3.3 and 3.4**
>
> Thank you for this suggestion. We agree that it would improve the clarity significantly. We will adjust the presentation of the results accordingly.
>
> **Q.4. the regret scales with multiplicative Lipschitz constants $G_f G_g$, from current analysis it does not seem to be possible to have additative dependence $G_f + G_g$, is this common?**
>
> Similar dependence on the Lipschitz constant is in the regret bound of prior work Mahdavi et al (2012). In particular, their Theorem 8 has the term $G^{5/2}$, where $G = \max\\{G_g, G_f\\}$.

---

### Official Review · Reviewer_yxBf · 2025-03-07

**Overall Recommendation:** 2

**Summary:**

This paper addresses the problem of Online Convex Optimization (OCO) with constraints, where the goal is to minimize regret while ensuring that the constraints are satisfied anytime. The authors propose an algorithm that combines gradient descent steps with Polyak feasibility steps to achieve $O(\sqrt{T})$ regret and anytime constraint satisfaction when the strict feasibility point is known. They also provide a guarantee when the strict feasibility point is not known. The authors validate their approach through numerical experiments, demonstrating that their algorithm achieves constraint satisfaction without sacrificing regret performance.

**Claims And Evidence:**

Yes, the claims are clear and convincing.

**Essential References Not Discussed:**

Yes, as discussed above [1] and [2].

**Experimental Designs Or Analyses:**

No specific issues.

**Methods And Evaluation Criteria:**

Given the vast literature on OCO with constraints, the paper might consider more baselines. For example, Yuan et al., 2018, could be a good baseline as they studied a strict violation metric.

**Other Comments Or Suggestions:**

Justifying the paper with [1] and [2] above and addressing the weakness above would be convincing. More baselines are needed in the experiments.

**Other Strengths And Weaknesses:**

Weaknesses:

- If I understand correctly, when the strictly feasibility does not hold as in Corollary 1, the algorithm would have a large period of $O(\log T)$ where the constraints could violate. This might not be proper for safety-critical applications.

- The paper considered the setting where the constraints are static. This seems less challenging and it would be more convincing to study stochastic or time-varying constraints.

**Questions For Authors:**

No further questions.

**Relation To Broader Scientific Literature:**

The contribution is marginal compared to the previous work.

My major concern is that the algorithm proposed in the paper has already appeared in previous work (e.g., [1] and [2]). [1] leverages the Polyak steps to deal with the stochastic constraints (which seems more challenging), and [2] already proposed the linear approximation in Algorithm 1 (SHAM), which is the same as the "First-order Approximation" in the paper.
The contribution is marginal.

[1] Ion Necoara and Nitesh Kumar Singh. Stochastic subgradient projection methods for composite
optimization with functional constraints. JMLR, 2022.

[2] Nitesh Kumar Singh and Ion Necoara. Stochastic halfspace approximation method for convex optimization with nonsmooth functional constraints. 2024.

**Theoretical Claims:**

I checked the theoretical claims and proofs, and they seem correct.

---

> ### Author Rebuttal · Authors · 2025-04-01
>
> We thank the reviewer for their detailed review. We have responded to each concern below.
>
> **Q1. The contribution is marginal compared to the previous work.**
>
> We respectfully disagree. The contribution of our paper is an algorithm for constrained OCO that *for the first time* attains anytime constraint satisfaction $g(x_t) \leq 0$ and optimal regret $O(\sqrt{T})$, while only using first-order constraint feedback. Unlike prior methods for this problem, which sacrifice feasibility for efficiency, our results show that it is possible to guarantee both feasibility and efficiency. Our approach is based on the use of Polyak steps with respect to the constraint, which has been long overlooked by the OCO literature.
>
> **Q2. My major concern is that the algorithm proposed in the paper has already appeared in previous work (e.g., [1] and [2]).**
>
> You are correct that similar algorithms have appeared in the offline optimization literature (see Section 1.1.3). However, the theoretical analysis used in the offline optimization literature is not applicable to the OCO setting. Specifically, the offline optimization works mentioned by the reviewer [1][2] *do not* provide applicable analysis for the following reasons:
> - [1],[2] require that the suboptimality gap is non-negative, i.e. $f(\hat{x}_t) - f^* \geq 0$, as used in the proofs for Theorem 7 in [1] and Theorem 4.6 [2]. In the online convex optimization setting, there is no guarantee that $f_t(x_t) - f_t(x^*) \geq 0$, and therefore we need a different analysis approach. See discussion on line 151 right side.
> - [1],[2] only give constraint violation guarantees on the average iterate $\hat{x} = \frac{1}{\sum_{\tau} \alpha_\tau} \sum_{\tau}  \alpha_\tau x_\tau$, whereas we provide violation guarantees on every iterate $x_t$, i.e. $g(x_t) \leq 0$ for all $t$, and therefore need a different analysis approach. Note that we *cannot* simply choose $x_t = \hat{x}_t$, because we only get feedback at $x_t$ and therefore the algorithm structure would no longer apply.
> - [1] requires constraint information on the intermediate iterate ($v_k$ in the notation of [1]), whereas we only use constraint information at the played actions $x_t$.
> - [2] requires that the cost function has Lipschitz gradients ($\beta$-smooth) as used in Lemma 4.1 of [2]. Since we do not assume Lipschitz cost gradients, we need a different analysis approach.
>
> Lastly, please note that [1] is already cited and discussed.
>
> **Q.3 [2] already proposed the linear approximation in Algorithm 1, which is the same as the "First-order Approximation" in the paper.**
>
> Thank you for pointing out that this part of our algorithm design is also used in [2], which studies (offline) optimization. We were not previously aware of this recent work. However, please note that the analysis from [2] does not apply to our setting for the reasons mentioned in the answer to Q.2 above. To clarify our contributions in the paper, we will add the following to Section 1.1.3.
>
> "This technique of using a first-order approximation is closely related to the approach used in the recent (offline) optimization work [2]. However, the analysis in [2] is not applicable to our OCO setting because (a) it requires that the suboptimality gap of the iterates $f( x_t ) - f^* \geq 0$ is non-negative, (b) only provides guarantees on the average of the iterates as opposed to our guarantees on every iterate, and (c) requires that the cost function has Lipschitz gradients."
>
> **Q4. When the strict-feasibility does not hold, the algorithm would have a large period of $O(\log(T))$ possible violation, which might not be proper for safety-critical applications.**
>
> Without knowledge of a feasible point, there is no way for any algorithm to ensure constraint satisfaction from the first round. Nonetheless, our results show that in such a setting, we can guarantee that the constraint is satisfied for all but a small number $O(\log(T))$ of rounds.
>
> In order to guarantee that the constraint is *always* satisfied, we require the slightly stronger assumption of a known *strictly-feasible* point. This assumption is motivated by the fact that, in safety critical-applications, there is typically a "safety margin" that is built in to traditional methods to account for modeling inaccuracies. For example, in personalized healthcare applications, medical standards specify a drug dosage that will be safe to the patient despite some small variations in the patient's characteristics (e.g. weight, blood pressure), and therefore this dosage can be used as a strictly-feasible point.
>
> **Q.5 The paper only considered the setting where the constraints are static.**
>
> The problem of OCO with fixed constraints is a major problem in the machine learning community. To illustrate this, Google Scholar shows that the first work on this problem (Mahdavi et al., 2012) has been cited 313 times.
>
> **More Experiments:** https://anonymous.4open.science/r/oco_pfs_rebuttal-1050/

---

### Official Review · Reviewer_3Pxe · 2025-03-10

**Overall Recommendation:** 2

**Summary:**

In this work, the authors study online convex optimization with a fixed constraint function. Their method employs Polyak feasibility steps to guarantee constraint satisfaction without compromising regret. Specifically, they introduce an algorithm for constrained OCO that applies Polyak feasibility steps to ensure anytime constraint satisfaction $g(x_t) \leq 0, \forall t$ and $\mathcal{O}(\sqrt{T})$ regret.

**Claims And Evidence:**

Yes, the claims are supported by the theorem.

**Essential References Not Discussed:**

The algorithm presented in this paper closely resembles the one proposed in [Liang, 2024] for Variational Inequality Problems with Functional Constraints and . It would be beneficial to discuss the similarities and differences between these approaches.

Reference:

[1]. Liang Zhang, Niao He, and Michael Muehlebach. Primal Methods for Variational Inequality Problems with Functional Constraints. 2024

**Experimental Designs Or Analyses:**

Yes. This paper only provides a toy example. It would be great to see some large-scale experiments.

**Methods And Evaluation Criteria:**

Yes, the proposed method makes sense for constrained online convex optimization problems.

**Other Comments Or Suggestions:**

What is $\alpha$ in line 314? Are you referring to $\epsilon$?

**Other Strengths And Weaknesses:**

Strength:
It is great to see polyak feasible stepsize is used in online convex optimization problems.

Weaknessses:
Assumption 4 is too strong, which is uncommon in prior literature [Yu, 2017], [Yuan, 2018]. It would be helpful to provide some justification and intuition on when this assumption holds.


References:

[1]. Hao Yu, Michael Neely, and Xiaohan Wei. Online convex optimization with stochastic constraints. Advances in Neural Information Processing Systems, 30, 2017.

[2]. Jianjun Yuan and Andrew Lamperski. Online convex optimization for cumulative constraints. Advances in Neural Information Processing Systems, 31, 2018.

**Questions For Authors:**

I am curious whether the tightening parameter $\rho$ is essential for this algorithm. It appears that setting $\rho = 0$ would still preserve most of the results. Could you clarify what would change if the tightening parameter $\rho$ were omitted, i.e., if we set $\rho = 0$?

**Relation To Broader Scientific Literature:**

The key contribution of using Polyak feasible step sizes is closely related to the constrained optimization literature.

**Theoretical Claims:**

Yes. There are some issues:
1. Issue with $\mathcal{X}_{\rho}$ when $g^* = \min_x g(x) = 0$: In this case, $\mathcal{X}{\rho} = \emptyset$, which would break the analysis in line 300.
2. The result in Theorem 1 heavily relies on the relationship between $\epsilon$ and $\sigma$. If $\frac{\epsilon}{\sigma}$ is very large, the bound will blow up. Could you provide results justifying that this ratio remains bounded?

---

> ### Author Rebuttal · Authors · 2025-04-01
>
> We thank the reviewer for their detailed review. We have responded to each concern below.
>
> **Q1. The algorithm in the paper closely resembles the one in Liang (2024), which studies constrained variational inequalities.**
>
> We were not previously aware of the recent work (Liang et al, 2024), which studies variational inequalities with functional constraints. Similar to our work, the algorithm in Liang et al (2024) uses a single iterate sequence and only uses constraint information at the chosen iterates. However, the analysis in Liang et al (2024) *cannot* be applied to our setting for the following reasons:
>
> 1. Liang et al (2024) assumes that the constraint functions $g_i$ has Lipschitz gradients (i.e. $\beta$-smooth), whereas we do not. This assumption is critical to the analysis in Liang et al. (2024), e.g. in equation (5), and therefore we require a different analysis approach.
> 2. Liang et al (2024) requires solving a QP subproblem at each iteration. On the other hand, our approach doesn't require solving a subproblem. This is critically important in the constrained OCO literature, because the constrained OCO problem was introduced for the sole purpose of avoiding subproblems and their computational burden. This is explicitly stated in Mahdavi et al. (2012), which introduced the constrained OCO problem.
> 3. Liang et al (2024) only studies variational inequalities, which *does not* extend to the OCO setting in general.
>
> Although our analysis is completely distinct from the one in Liang et al (2024), there are some similarities in the *algorithm design* for the special case of fixed cost functions $f_t = f$, one constraint function, and  $\beta$-smooth constraints. In this case, our algorithm can be written as (with $\rho = 0$),
> $$
> x_{t+1} = \Pi_{R \mathbb{B}} \left( x_t - \eta \nabla f(x_t) - \frac{[g(x_t) - \eta \nabla f(x_t)^\top \nabla g(x_t)]_+}{|| \nabla g(x_t) ||^2} \nabla g(x_t) \right).
> $$
> At a high-level, this resembles the version of the algorithm in Liang et al. (2024) for the special case of one constraint function (see Algorithm 2 in Liang et al. (2024)).
> However, there are some key differences:
> - Liang et al. (2024) applies the decreasing step-size $\eta$ to the entire third term, whereas we only apply it to $\nabla f(x_t)^\top \nabla g(x_t)$.
> - Liang et al. (2024) applies the scaling $\alpha = G_g/R$ to $g(x_t)$.
> - Liang et al. (2024) adds an additional constraint to ensure that the iterates are bounded, whereas we use a projection onto $R \mathbb{B}$.
>
> These differences in algorithm design further highlight the needed for distinct analysis between our work and Liang et al. (2024).
>
> We thank the reviewer for pointing out this work and will add the above discussion to the paper.
>
> Liang et al. "Primal Methods for Variational Inequality Problems with Functional Constraints," arXiv:2403.12859
>
> **Q2. Assumption 4 is too strong and uncommon in the prior literature.**
>
>
> We respectfully disagree with the statement that Assumption 4 is uncommon in prior literature, as it is used in the fundamental works on OCO with constraints (Mahdavi et al., 2012; Jennaton et al. 2016). To help put this assumption in context, we note that it is implied by Slater's condition (provided that the other assumptions hold).
>
> To see this, suppose that Slater's condition holds, i.e. there exists $y \in \mathbb{R}^d$ and $\xi > 0$ such that $g(y) \leq - \xi$. Then, let $\epsilon = \xi/2$ and $\sigma = \xi/(4R)$. Then, it holds for any $x \in \mathbb{R}^d$ and $s \in \partial g(x)$ where $g(x) = -\epsilon$,
> $$
> -\xi \geq g(y) \geq g(x) + s^\top(y- x) \geq - \epsilon - || s || || y - x || \geq - \epsilon - 2 || s || R.\\
> $$
> $$
> \implies || s || \geq \frac{\xi - \epsilon}{2 R} =  \frac{\xi}{4 R} = \sigma.
> $$
> This is precisely Assumption 4.
>
> **Q3. There is an issue with the analysis in that it is possible for $\mathcal{X}_\rho = \emptyset$.**
>
> Thank you for pointing this out. This is due to a typo in Assumption 4. There is a missing negative, the $g(x) =\epsilon$ is supposed to be $g(x) = - \epsilon$. The claims of the paper remain unchanged by fixing this typo, as the correct form of the assumption is used throughout the analysis.
>
> **Q.4. The result in Theorem 1 heavily relies on the relationship between $\epsilon$ and $\sigma$. Could you provide results justifying that the ratio $\frac{\epsilon}{\sigma}$ remains bounded?**
>
> Similar to prior work (Mahdavi et al., 2012; Jennaton et al., 2016), the terms $\epsilon$ and $\sigma$ are assumed to be constants and the regret bound depends on them.
>
> **Q.5. Could you clarify what would change if the tightening parameter were omitted, i.e., if we set $\rho=0$?**
>
> We kindly point the reviewer to Corollary 2 in the paper. This corollary shows that choosing $\rho = 0$ ensures that $g(x_t) \leq O(1/\sqrt{t})$.
>
> **Q.6. What is $\alpha$ in line 314? Are you referring to $\epsilon$?**
>
> Thank you for pointing out this typo. It is supposed to be $\rho$ instead of $\alpha$.

---

### Decision · Program_Chairs · 2025-05-01

**Decision:**

Accept (poster)

**Comment:**

The paper presents a $O(\sqrt{T})$ regret analysis for online constrained problems which also guarantees that the constraint $g(x_t) \leq 0$ holds any time for the iterates $x_t$. This is the first such regret analysis to guarantee that the constraint holds (as opposed to a regret over the constraint).
The method underlying this analysis relies on linearizing the constraint, and using the standard gradient model for the objective. The reviewers have raised that this is very similar to a method used in offline optimization [1] and [2], and that the analysis might also be similar.  Because the reviewers were not overall favorable, I have gone into the details of this paper and the papers that the reviewers find similar [1] and [2]. The similarity in the method is not surprising, this is a variant of one of the most standard SQP type methods. What stands out in this paper is identifying that exactly this variant of the method, together with a carefully constructed analysis could achieve the correct regret and any time feasibility. The analysis in [1] and [2] does not carry over for a number of reasons (including using stationarity conditions that do not hold online). As such, this paper solves an interesting  valid open problem in online learning. Which is why I recommend it be accepted.


[1] Liang Zhang, Niao He, and Michael Muehlebach. Primal Methods for Variational Inequality Problems with Functional Constraints. 2024

[2] Nitesh Kumar Singh and Ion Necoara. Stochastic halfspace approximation method for convex optimization with nonsmooth functional constraints. 2024.